# WristWorld: Generating Wrist-Views via 4D World Models for Robotic Manipulation

## Abstract

Wrist-view observations are crucial for VLA models as they capture fine-grained hand–object interactions that directly enhance manipulation performance. Yet large-scale datasets rarely include such recordings, resulting in a substantial gap between abundant anchor views and scarce wrist views. Existing world models cannot bridge this gap, as they require a wrist-view first frame and thus fail to generate wrist-view videos from anchor views alone. Amid this gap, recent visual geometry models such as VGGT emerge with precisely the geometric and cross-view priors that make it possible to address such extreme viewpoint shifts. Inspired by these insights, we propose **WristWorld**, the first 4D world model that generates wrist-view videos solely from anchor views. WristWorld operates in two stages: (i) *Reconstruction*, which extends VGGT and incorporates our proposed Spatial Projection Consistency (SPC) Loss to estimate geometrically consistent wrist-view poses and 4D point clouds; (ii) *Generation*, which employs our designed video generation model to synthesize temporally coherent wrist-view videos from the reconstructed perspective. Experiments on Droid, Calvin, and Franka Panda demonstrate state-of-the-art video generation with superior spatial consistency, while also improving VLA performance, raising the average task completion length on Calvin by 3.81% and closing **42.4%** of the anchor-wrist view gap. See video results at anonymous page: *wrist-world.github.io*

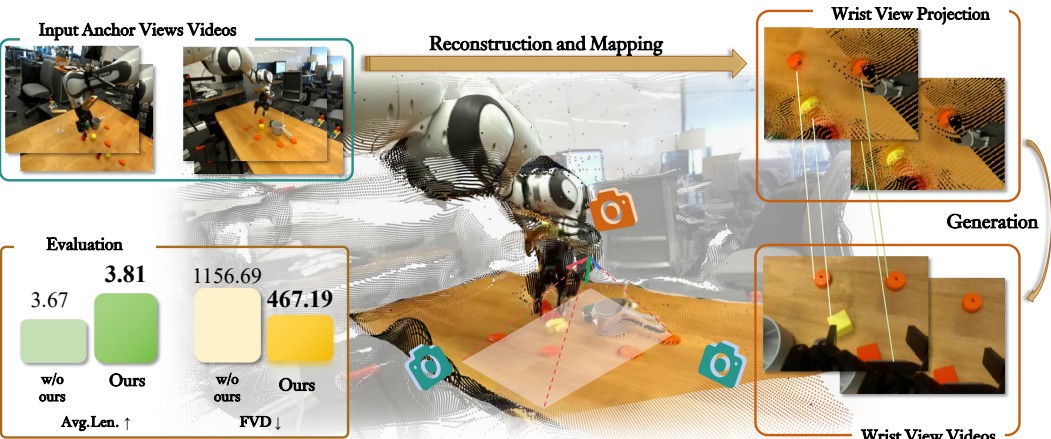

Figure 1: We present **WristWorld**, a framework that synthesizes realistic wrist-view videos from anchor views through a two-stage process: a *reconstruction stage* for estimating wrist-view projections, and a *generation stage* for producing coherent wrist-view videos. The generated wrist observations effectively expanding training data to novel view and lead to significant performance improvements for downstream VLA models across various tasks.

# 1 INTRODUCTION

Wrist-view observations play a central role in vision–language–action (VLA) models because they directly capture the fine-grained hand–object interactions that underlie precise manipulation. However, most large-scale robotic datasets provide only limited wrist-view coverage (Ebert et al., 2024; Mandlekar et al., 2018; Collaboration et al., 2025), producing a substantial and practically important gap between abundant anchor (third-person) views and scarce wrist-centric recordings. Models pretrained on external perspectives therefore often underperform when tasks require detailed wrist-centric perception or control.

Collecting wrist-view data at scale is expensive: it demands extra sensors, careful calibration, and specialized recording setups. While world models have been proposed for data completion and synthesis (Wang et al., 2025c; Yang et al., 2023; Zhen et al., 2025), existing approaches are not designed to close the anchor-to-wrist gap in realistic, dynamic manipulation settings. Many of them (Liao et al., 2025; Liu et al., 2024) require a wrist-view first frame as a condition and thus cannot generate wrist-view sequences from anchor views alone. This raises a natural question: ***can we enrich existing third-person datasets with automatically generated, geometrically consistent wrist-view sequences that support both perception and control?***

However, bridging anchor views to wrist views is highly challenging: First, scenes are dynamic and dominated by articulated arms and manipulators that cause severe, time-varying occlusions. Second, the target wrist perspective is often not seen during training of current viewpoint transfer methods. Third, geometric reconstructions from anchor views are typically sparse and temporally inconsistent, making naïve view-synthesis prone to spatial or temporal artifacts (Liu et al., 2024).

Motivated by recent progress in visual geometric models (Wang et al., 2025b)) and diffusion-based video synthesis (Blattmann et al., 2023), we present **WristWorld**, the *4D* world model that synthesizes wrist-view videos solely from anchor views. WristWorld is a two-stage pipeline that explicitly enforces geometric and temporal consistency. In the *Reconstruction* stage we extend VGGT with a dedicated wrist head that encodes the extreme viewpoint transform and estimates geometrically consistent 4D point clouds and wrist-view camera poses. A novel *Spatial Projection Consistency* (SPC) loss supervises alignment between 2D correspondences and the reconstructed 3D/4D geometry to improve spatial fidelity. In the *Generation* stage, a diffusion-based video generator conditioned on the reconstructed wrist projections and CLIP-encoded anchor-view features synthesizes temporally coherent wrist-view videos that respect the recovered geometry and scene semantics.

We validate WristWorld on Droid (Khazatsky et al., 2024), Calvin (Mees et al., 2022), and Franka Panda setups. Results show state-of-the-art wrist-view video generation with superior spatial consistency, and practical downstream gains for VLA: on Calvin we increase average task completion length by **3.81%** and close **42.4%** of the anchor–wrist performance gap. Importantly, WristWorld can be used as a plug-in to extend existing single-view world models with multi-view capability without requiring new wrist-view data collection. **Our contributions are three-fold:**

- A novel two-stage framework for anchor-view to wrist-view video generation that achieves both temporal Consistency and geometric consistency.

- Leveraging a wrist head, SPC loss, and CLIP-encoded anchor-view features to synthesize consistent wrist-view sequences from anchor views.

- Experiments showing that our approach improves VLA performance and can be applied in a plug-and-play manner to extend single-view world models into multi-view settings.

# 2 RELATED WORK

**3D Reconstruction for Robotic Perception**. Accurate multi-view 3D reconstruction and camera pose are key for manipulation, yet many frameworks assume fixed calibration or static views. Recent work injects geometry into policies; e.g., GNFactor jointly optimizes a NeRF scene model and a manipulation policy, sharing one 3D representation for multi-task learning (Ze et al., 2023). Transformer-based vision models have also been explored; for instance, VGGT encodes multi-view observations into fused geometric features for 3D prediction (Wang et al., 2025b). Still, moving wrist-camera pose is rarely modeled, and large datasets rely on manual calibration instead of online

wrist-centric pose prediction. Cross-view consistency remains crucial in dynamic 3D scene reconstruction (Wang et al., 2025d; Hu et al., 2025), with MTVCrafter introducing 4D motion tokens to enforce coherence (Ding et al., 2025).

**Video Generation Models for Manipulation**. Diffusion-based video generators let planners "imagine" robot futures. Web-scale text-to-video models look realistic but struggle with novel object–action combinations. RoboDreamer improves compositionality by factorizing generation via language parsing (Zhou et al., 2024). This&That adds gesture conditioning for controllable plans beyond text-only inputs (Wang et al., 2025a). Coupling prediction with control, VideoAgent iteratively self-refines diffused plans to reduce hallucinations (Soni et al.), and action-conditioned diffusion in generative predictive control approximates dynamics for policy improvement (Qi et al., 2025). Synthetic data routes like DreamGen synthesize diverse "dream" trajectories for stronger generalization (Jang et al., 2025). For spatial consistency, EnerVerse combines multi-view diffusion with 4D reconstruction (Gaussian splatting) to produce geometry-consistent futures and better long-horizon planning (Huang et al., 2025; Li et al., 2025). Large frameworks such as UniPi use text-guided video generation to learn universal multi-task policies (Du et al., 2023; Chi et al.).

**Vision–Language–Action (VLA) Robotics Models**. VLA policies learn directly from paired visual and linguistic inputs without constructing an explicit world model. GR-1, a GPT-style video-conditioned policy, is pretrained on large human video corpora and fine-tuned to achieve state-of-the-art multi-task performance on CALVIN (88.9% to 94.9%) with zero-shot generalization to novel scenes (Wu et al., 2023; Mees et al., 2022). GR-2 scales training to 38M video–text pairs, producing a generalist agent capable of executing 100+ manipulation tasks by grounding instructions in action sequences (Cheang et al., 2024). Beyond internet video pretraining, Vid2Robot maps human video demonstrations to robot policies via cross-attention (Jain et al., 2024), and MimicPlay derives hierarchical plans from unstructured human play (Wang et al., 2023). Label-free alignment has also been explored by grounding a frozen video generator into continuous actions through goal-conditioned self-exploration (Luo & Du, 2024). Conversely, human-in-the-loop fine-tuning attains high dexterity yet forgoes an explicit visual world model (Luo et al., 2025; Chi et al., 2025).

# 3 METHOD

Our method is a two-stage 4D Generative World Model designed to synthesize geometrically consistent wrist-view videos from third-person observations. The first *reconstruction stage* estimates wrist poses and generates condition maps via point cloud projection. The second *generation stage* synthesizes temporally coherent wrist-view sequences conditioned on these maps and enriched by semantic guidance. The overall framework is illustrated in Figure 2.

## 3.1 PRELIMINARY

**Video Diffusion Models.** Recent advances in video synthesis are largely driven by diffusion-based generative models. A video $\mathbf{X} = \{x^t\}_{t=1}^T$ is first compressed into a latent representation $\mathbf{Z}_0 = \{z^t\}_{t=1}^T \in \mathbb{R}^{T \times C \times H \times W}$ using a video VAE, which reduces spatial and temporal resolution while preserving content semantics. The diffusion framework then defines a forward noising process that gradually perturbs $\mathbf{Z}_0$ into Gaussian noise, and a learned denoising model $\epsilon_\theta$ that reverses this process. At training time, the objective is to predict the added noise given the noisy latent $\mathbf{Z}_\tau$ at step $\tau$ and the conditioning signal $\mathbf{c}$:

$$\mathcal{L}_{\text{diff}} = \mathbb{E}_{\mathbf{Z}_0, \, \epsilon, \, \tau} \left\| \epsilon - \epsilon_\theta(\mathbf{Z}_\tau, \tau \mid \mathbf{c}) \right\|_2^2,$$

where $\epsilon \sim \mathcal{N}(0, \mathbf{I})$ and $\mathbf{Z}_\tau$ is obtained by a variance-scheduled corruption of $\mathbf{Z}_0$. In a Diffusion Transformer (DiT), conditioning $c$ is typically realized as *text embeddings*, which are projected into conditioning tokens and injected into the transformer blocks, thereby guiding the denoising process.

**Visual Geometry Models.** To capture multi-view geometry and establish dense cross-view correspondences, we build on VGGT (Wang et al., 2025b), a large Transformer that encodes multi-camera observations into fused features $\mathbf{F}$ and predicts 3D quantities such as point clouds and correspondences. Given a query point $\mathbf{u}_q^j$ in image $I_q$, the matching head predicts corresponding points $\{\hat{\mathbf{u}}_i^j\}_{i=1}^N$ in other views $\{I_i\}$, where $N$ is the number of anchor views and $M$ the number of sampled

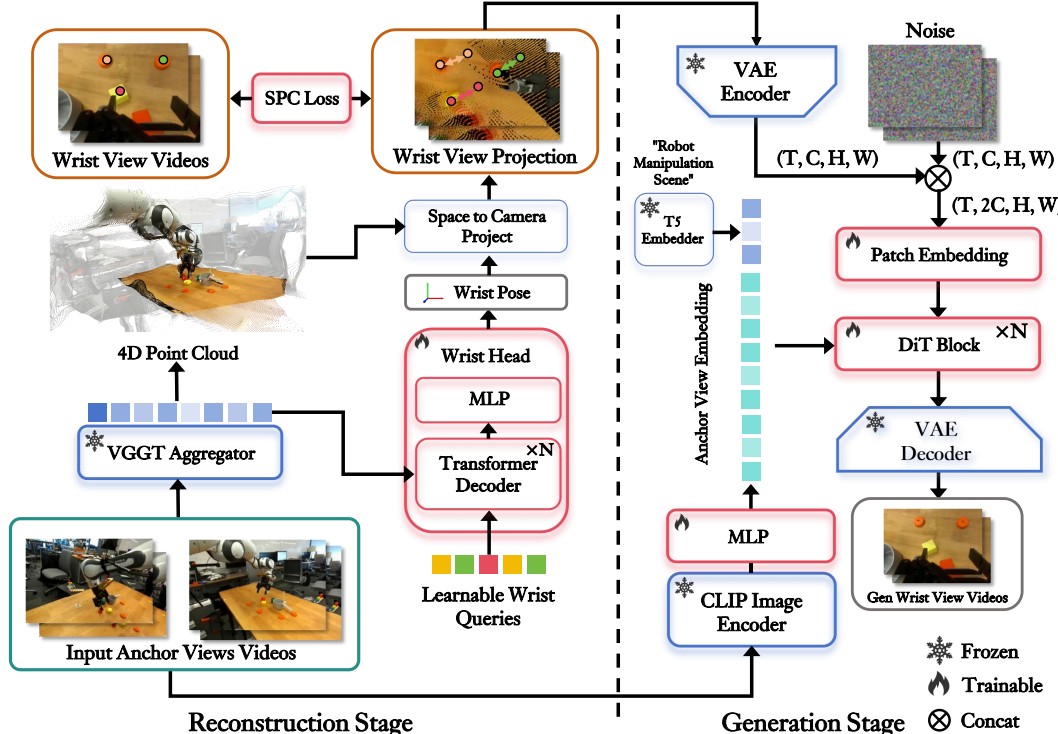

Figure 2: **Overview of our method.** We introduce a two-stage 4D Generative World Model. In the reconstruction stage, VGGT Wang et al. (2025b) is extended with a wrist head to regress wrist pose, guided by a Spatial Projection Consistency Loss that supervises directly from RGB without depth or extrinsics. The predicted pose projects point clouds into the wrist view. In the generation stage, these projections, combined with external-view CLIP embeddings, condition a video generator to synthesize wrist-view sequences. Without first-frame guidance, the model produces additional wrist views for VLA datasets, yielding substantial performance gains.

query points. The resulting correspondences are

$$\mathcal{C} = \{(\mathbf{u}_q^j, \hat{\mathbf{u}}_i^j)\}_{i=1,\dots,N}^{j=1,\dots,M},$$

providing dense pixel-level matching across anchor and wrist views. We adopt the pinhole camera model, where a 3D point $\hat{\mathbf{y}} \in \mathbb{R}^3$ projects to pixel $\mathbf{u} \in \mathbb{R}^2$ via camera intrinsics $\mathbf{K}$, extrinsics $(\mathbf{R}, \mathbf{T})$, and projection function $\Pi(\cdot)$:

$$\mathbf{u} = \Pi(\mathbf{K}, \mathbf{R}, \mathbf{T}, \hat{\mathbf{y}}).$$

### 3.2 RECONSTRUCTION STAGE

**Wrist Head Design.** To estimate the wrist-mounted viewpoint, we extend VGGT with a specialized *wrist head*. Based on the aggregated multi-view features $\mathbf{F}$, we introduce a set of learnable wrist queries that attend to these tokens through a Transformer decoder. The wrist head regresses the wrist camera extrinsics, denoted as rotation $\mathbf{R}_w \in SO(3)$ and translation $\mathbf{T}_w \in \mathbb{R}^3$:

$$(\mathbf{R}_w, \mathbf{T}_w) = \text{WristHead}(\mathbf{F}, \mathbf{q}_w),$$

where $\mathbf{q}_w$ are the wrist queries. This design allows the model to capture hand-centered motion and implicit camera pose even when wrist-view data is unavailable.

**Spatial Projection Consistency Loss.** Direct supervision of wrist extrinsics or depth maps is often missing. To address this, we propose a *Spatial Projection Consistency (SPC) loss* that enforces geometric consistency from RGB correspondences alone. As shown in Figure 3, given dense 2D–2D correspondences $\mathcal{C} = \{(\mathbf{u}_q^j, \hat{\mathbf{u}}_w^j)\}_{j=1}^M$ between an anchor view $I_q$ and the wrist view $I_w$, and a

reconstructed point cloud $\mathcal{Y} = \{\hat{\mathbf{y}}_j\}$ from anchor views, we associate each anchor pixel $\mathbf{u}_q^j$ with its corresponding 3D point $\hat{\mathbf{y}}_j \in \mathcal{Y}$. This yields a set of 3D–2D pairs

$$\mathcal{T} = \{(\hat{\mathbf{y}}_j, \hat{\mathbf{u}}_w^j)\}_{j=1}^M,$$

linking reconstructed world points to wrist-view pixels.

For each pair $(\hat{\mathbf{y}}_j, \hat{\mathbf{u}}_w^j)$, the projection of $\hat{\mathbf{y}}_j$ under the predicted wrist pose $(\mathbf{R}_w, \mathbf{T}_w)$ is denoted by $\mathbf{u}_w'^j = \Pi(\mathbf{K}, \mathbf{R}_w, \mathbf{T}_w, \hat{\mathbf{y}}_j)$, where $\mathbf{K}$ is fixed by the dataset. We then divide points into $\mathcal{S}_{\text{front}}$ for positive depth values and $\mathcal{S}_{\text{back}}$ for negative ones. The SPC loss consists of two terms:

$$\mathcal{L}_u = \frac{1}{|\mathcal{S}_{\text{front}}|} \sum_{\hat{\mathbf{y}}_j \in \mathcal{S}_{\text{front}}} \text{MSE}(\mathbf{u}_w'^j, \hat{\mathbf{u}}_w^j), \qquad \mathcal{L}_{\text{depth}} = -\frac{1}{|\mathcal{S}_{\text{back}}|} \sum_{\hat{\mathbf{y}}_j \in \mathcal{S}_{\text{back}}} z_j.$$

where $z_j$ is the depth value of point $\hat{\mathbf{y}}_j$ in the wrist camera coordinate frame. Finally, the projection loss is defined as $\mathcal{L}_{\text{proj}} = \lambda_u \mathcal{L}_u + \lambda_{\text{depth}} \mathcal{L}_{\text{depth}}$, where $\lambda_u$ and $\lambda_{\text{depth}}$ control the balance between reprojection consistency and depth feasibility.

**Condition Map Generation.** With the estimated wrist poses across frames, reconstructed 3D scenes are projected into the wrist-view image plane to form a temporally aligned sequence of *condition maps*. These maps provide frame-consistent structural guidance for the subsequent video generation stage.

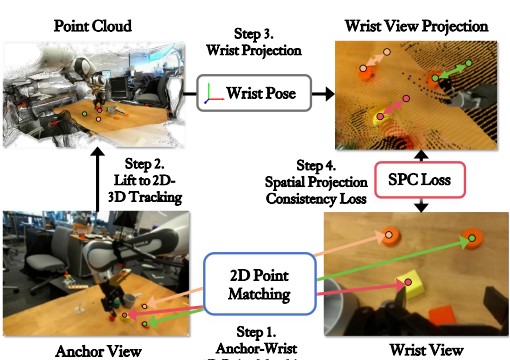

### 3.3 GENERATION STAGE

**Video Generation Model.** We adopt a DiT (Peebles & Xie, 2022) for video synthesis and introduce two targeted modifications. First, the conditioning $\mathbf{c}$ comprises third-person CLIP features together with text embeddings to modulate the DiT. Second, we modify the patch embedding to ingest the concatenated latent stream $\mathbf{Z}_0 = \{[\mathbf{z}_w^t; \mathbf{z}_c^t]\}_{t=1}^T \in \mathbb{R}^{T \times 2C \times H \times W}$, expanding the standard input from $(T, C, H, W)$ to $(T, 2C, H, W)$.

Figure 3: **Spatial Projection Consistency (SPC) loss.** We first establish anchor–wrist 2D point matching and then lift the matched pixels to 2D–3D correspondences using the reconstructed point cloud. The 3D points are subsequently projected into the wrist view with the predicted wrist pose, after which the SPC loss is computed to enforce geometric consistency.

**Wrist-View-Projection-Guided Generation.** Since the projected condition maps are geometrically aligned with the wrist viewpoint, they directly provide spatial structure. We encode each wrist view projection $\mathbf{C}^t$ into a latent representation $\mathbf{z}_c^t$ using a VAE, and concatenate it with the noisy wrist latent $\mathbf{z}_w^t$:

$$\mathbf{z}^t = [\mathbf{z}_w^t; \mathbf{z}_c^t],$$

which is then processed by the video generation model. This integration allows the model to synthesize temporally coherent wrist-view sequences that remain consistent with 3D geometry.

**CLIP-Encoded Anchor-View Semantics.** Because condition maps may miss global semantics (e.g., small or blurred objects from point-cloud projection), we introduce an external semantic pathway. Each third-person frame from $N$ anchor views is encoded by a CLIP image encoder to obtain per-frame, per-view features:

$$\mathbf{E}_{\text{clip}} = \{\mathbf{e}_{t,i}\}_{i=1,\dots,N;\ t=1,\dots,T} \in \mathbb{R}^{(NT) \times d_c},$$

where $\mathbf{e}_{t,i}$ denotes the CLIP embedding of the $t$-th frame from the $i$-th anchor view.

A text prompt is also encoded, and both modalities are projected into a shared conditioning space:

$$\tilde{\mathbf{E}}_{\text{clip}} = W_c \mathbf{E}_{\text{clip}}, \qquad \tilde{\mathbf{E}}_{\text{text}} = W_t \mathbf{E}_{\text{text}}.$$

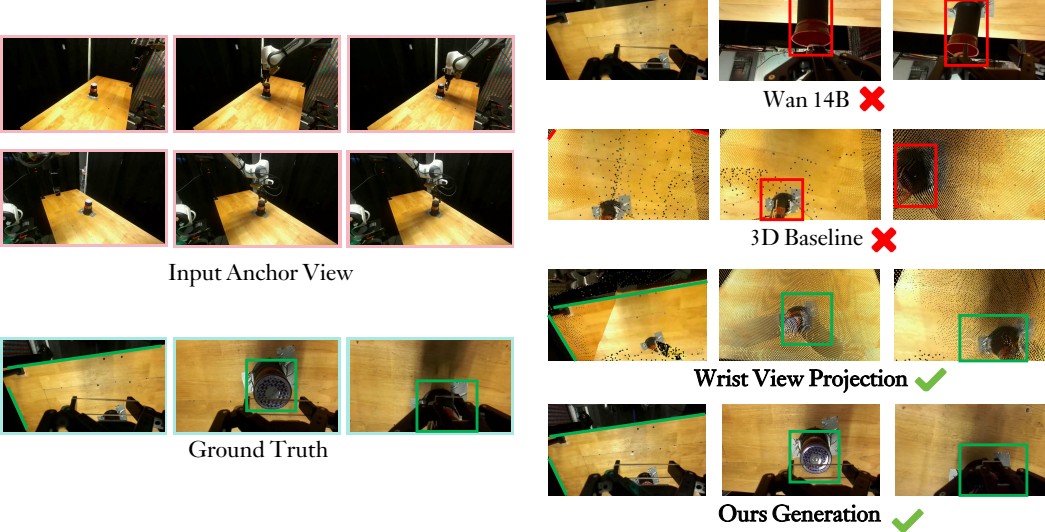

Figure 4: **Visualization of our generation result.** As illustrated in the figure, we compare our generated condition maps against the 3D Base (VGGT without the SPC Loss), where our approach demonstrates superior viewpoint consistency. Furthermore, in comparison to the Wan14B (Wan et al., 2025) baseline, our method achieves both higher generation quality and improved viewpoint alignment accuracy. These results highlight the effectiveness of our framework and underscore its potential to serve as training data for downstream VLA models.

| Method | Wrist RGB First Frame | Droid | | | | Franka Panda | | | |
|---|---|---|---|---|---|---|---|---|---|
| | | FVD↓ | LPIPS↓ | SSIM↑ | PSNR↑ | FVD↓ | LPIPS↓ | SSIM↑ | PSNR↑ |
| VGGT (Wang et al., 2025b) | × | - | 0.74 | 0.28 | 9.56 | - | 0.73 | 0.49 | 12.05 |
| Pix2Pix (Isola et al., 2017) | × | - | 0.55 | 0.58 | 12.81 | - | 0.58 | 0.71 | 15.60 |
| Wan 1.3B (Wan et al., 2025) | × | 1142.15 | 0.61 | 0.46 | 10.08 | 1944.59 | 0.72 | 0.53 | 10.45 |
| SVD (Blattmann et al., 2023) | ✓ | 2005.44 | 0.56 | 0.50 | 11.12 | 1354.56 | 0.60 | 0.68 | 14.10 |
| Cosmos-Predict2 (NVIDIA, 2025) | ✓ | 1990.72 | 0.51 | 0.56 | 12.74 | 1156.69 | 0.65 | 0.67 | 12.59 |
| Wan 14B (Wan et al., 2025) | ✓ | 935.03 | 0.53 | 0.54 | 11.98 | 985.99 | 0.59 | 0.68 | 13.93 |
| **Ours** | × | **421.10** | **0.39** | **0.64** | **14.78** | **231.43** | **0.33** | **0.78** | **17.84** |

Table 1: Quantitative comparison on Droid and our Franka Panda. Green rows denote methods without wrist-view input, yellow rows require a wrist-view first frame, and red highlights our method. Our method achieves the best performance across all metrics without first-frame guidance.

We then form the conditioning tokens by concatenating along the token dimension:

$$\mathbf{c} = \left[ \tilde{\mathbf{E}}_{\text{clip}} + \mathbf{p}_{\text{temporal}}^{1:T} + \mathbf{p}_{\text{view}}^{1:N} \; ; \; \tilde{\mathbf{E}}_{\text{text}} + \mathbf{p}_{\text{text}} \right],$$

where $\mathbf{p}_{\text{temporal}}^{1:T}$ are temporal embeddings, $\mathbf{p}_{\text{view}}^{1:N}$ are view-identity embeddings distinguishing the $N$ anchor views, which are both learnable parameters. And $\mathbf{p}_{\text{text}}$ is a text-token positional embedding. The resulting $\mathbf{c}$ injects global semantics into the video generation process.

## 4 EXPERIMENT

### 4.1 IMPLEMENTATION DETAILS

**Dataset.** We conduct experiments on three sources of data:

1. **Droid (Khazatsky et al., 2024).** The *Droid* dataset is a large-scale robotics video corpus with about 76k videos covering 59 diverse manipulation tasks. Each video is recorded at 1280×720 resolution from multiple static viewpoints, including *ext1*, *ext2*, and a wrist-mounted camera. For pretraining, we sample a 10k subset and use two anchor views as model inputs. For evaluation, we additionally hold out 100 videos as a validation set.

| Method | Inputs | 1/5 | 2/5 | 3/5 | 4/5 | 5/5 | Avg. Len. |
|---|---|---|---|---|---|---|---|
| MDT (Reuss et al., 2024) | Static RGB + Gripper RGB | 93.7% | 84.5% | 74.1% | 64.4% | 55.6% | 3.72 |
| HULC++ (Mees et al., 2023) | Static RGB + Gripper RGB | 93% | 79% | 64% | 52% | 40% | 3.30 |
| VPP (Hu et al., 2024) | Static RGB + Gripper RGB | 94.9% | 86.8% | 80.4% | 72.9% | 65.4% | 4.00 |
| SuSIE (Black et al., 2023) | Static RGB | 87.7% | 67.4% | 49.8% | 41.9% | 33.7% | 2.80 |
| TaKSIE (Kang & Kuo, 2025) | Static RGB | 90.4% | 73.9% | 61.7% | 51.2% | 40.8% | 3.18 |
| VPP + VGGT (Wang et al., 2025b) | Static RGB | 91.8% | 79.9% | 65.4% | 54.3% | 44.1% | 3.33 |
| VPP | Static RGB | 91.2% | 82.2% | 73.2% | 65.2% | 55.4% | 3.67 |
| **VPP + Ours** | Static RGB | **92.9%** ↑1.7 | **84.2%** ↑2.0 | **75.4%** ↑2.2 | **67.6%** ↑2.4 | **60.4%** ↑5.0 | **3.81** ↑0.14 |

Table 2: VLA performance on the Calvin (Mees et al., 2022) benchmark with and without wrist-view generation. We use the Video Prediction Policy (VPP) (Hu et al., 2024) as our VLA. In Calvin, each episode consists of five sequential tasks and terminates once a failure occurs. The columns 1/5–5/5 report success rates of completing at least $k$ tasks in sequence, while *Avg. Len.* denotes the mean number of tasks completed per episode.

| Inputs | Close the upper drawer | Pick bread and place into drawer | Pick up the milk | Mean |
|---|---|---|---|---|
| Anchor + Wrist | 80.0% | 73.3% | 46.7% | 66.7% |
| Anchor | 60.0% | 40.0% | 13.3% | 37.8% |
| **Anchor + Ours Gen Wrist** | 73.3% ↑13.3 | 53.3% ↑13.3 | 33.3% ↑20.0 | 53.3% ↑15.5 |

Table 3: VLA performance on our Franka Panda dataset with and without wrist-view generation.

2. **Calvin (Mees et al., 2022).** To benchmark vision-language-action learning in simulation, we adopt the *Calvin* environment. Calvin provides multi-view demonstrations across multiple task splits; in this work we focus on the D split and use 10% of the data for training. For downstream VLA models, we follow the standard Task $D \rightarrow D$ configuration for training and evaluation.

3. **Franka Panda.** Beyond simulation, we collect 1700 demonstrations on a real Franka Panda manipulator. Our setup includes three static cameras (*left*, *right*, *top*) and one wrist-view camera. Videos are captured at 30 fps and downsampled temporally by a factor of three. For evaluation, we hold out 100 videos from this collection.

**Training.** Our framework is trained in two stages: reconstruction and video generation. During pretraining on the Droid dataset, the reconstruction stage is optimized on 8×A800 GPUs for 12 hours with a batch size of 4 and resolution of 640×480, followed by the video generation stage on 8×A800 GPUs for 24 hours using a condition token length of 512. Building upon this, we perform cross-view fine-tuning with the Franka demonstrations. Since migrating from two views to three changes the dimensionality of the view-identity embeddings $p_{view}$, our cross-view fine-tuning loads all pretrained weights while randomly re-initializing only the view-embedding parameters, which are then jointly optimized during fine-tuning. The reconstruction stage fine-tuning requires 8 GPUs for 6 hours, while the generation stage is trained on 8 GPUs for 12 hours under the same batch size, resolution, and token length settings.

**Model.** Our framework consists of two stages: reconstruction and generation. In the *reconstruction stage*, a VGGT backbone encodes multi-view features, and a dedicated *wrist head* predicts wrist camera parameters through attention-based token fusion and transformer refinement, supervised by projection and optional L1 losses. In the *generation stage*, We use the VAE (Kingma & Welling, 2013) compresses frames into latent sequences, while the DiT applies spatio-temporal attention on tokenized latents. Conditioned jointly on text and visual features, the DiT generates geometrically consistent and temporally coherent videos.

### 4.2 Video Generation Quantitative Evaluation

We compare against recent baselines on Droid and Franka-Panda (Tab. 1). Our method surpasses prior work across all metrics (FVD (Unterthiner et al., 2019), LPIPS (Zhang et al., 2018), SSIM (Wang et al., 2004), PSNR) without first-frame guidance, achieving large FVD gains (temporal coherence) and improved perceptual and structural fidelity. Even on the challenging Franka-Panda dataset with viewpoint variation, our framework consistently outperforms all baselines.

Qualitative results are shown in Fig. 4, Fig. 5, and Fig. 6. In Fig. 4, our wrist view projection yield more viewpoint-aligned generations than both the 3D Base and Wan 14B. On Calvin (Fig. 5), our

| Wrist view Projection | Ext view clip embedding | SPC Loss | FVD↓ | LPIPS↓ | SSIM↑ | PSNR↑ |
|:---:|:---:|:---:|:---:|:---:|:---:|:---:|
| ✗ | ✓ | ✗ | 3091.74 | 0.74 | 0.55 | 10.42 |
| ✓ | ✓ | ✗ | 790.10 | 0.59 | 0.47 | 10.75 |
| ✓ | ✗ | ✓ | 474.32 | 0.44 | 0.61 | 13.67 |
| ✓ | ✓ | ✓ | **421.10** | **0.39** | **0.64** | **14.78** |

Table 4: Ablation on wrist view projection, clip embeddings, and SPC loss.

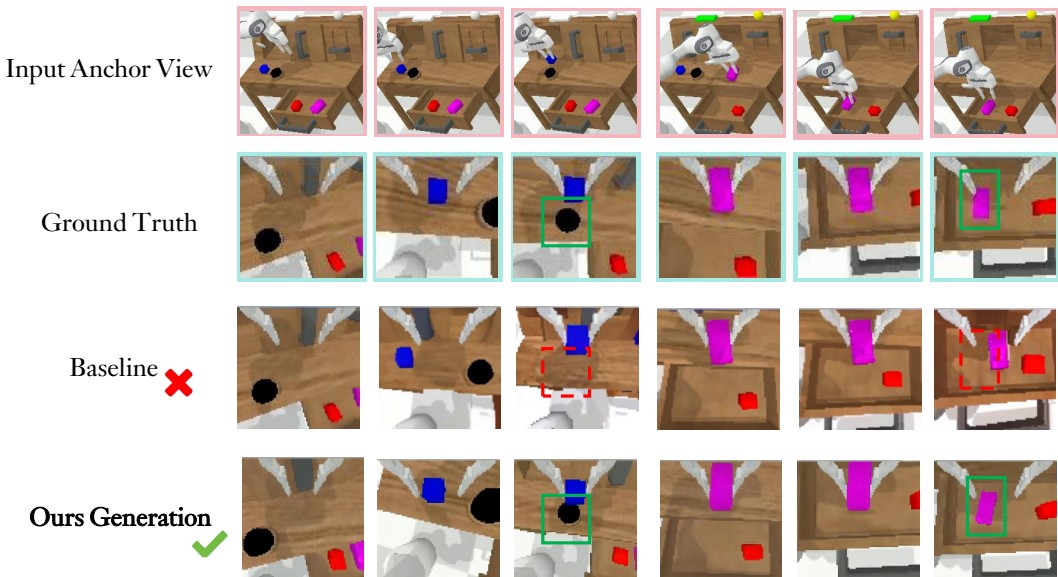

Figure 5: **Visualization on the Calvin (Mees et al., 2022) benchmark.** We compare our generated wrist-view videos (bottom row) with the ground truth (second row) and a baseline method (third row, Stable Video Diffusion (Blattmann et al., 2023)). Our approach achieves better spatial and viewpoint consistency than the baseline, while also producing more faithful wrist-view frames. These results highlight the effectiveness of our method in bridging anchor-view and wrist-view perspectives.

approach improves spatial/viewpoint consistency over SVD. On Franka Panda data (Fig. 6), our wrist-view generations closely match ground truth, showing strong third-to-wrist generalization.

Together, these qualitative and quantitative results highlight that our generated videos not only serve as high-quality reconstructions but also provide valuable data for downstream VLA models.

### 4.3 DATA-DRIVEN VLA ENHANCEMENT

We evaluate whether synthesized wrist-view videos improve vision–language–action (VLA) policies. Our framework generates wrist-view sequences from *anchor-view* rollouts and augments the training data of an unchanged VLA model (Video Prediction Policy, VPP (Hu et al., 2024)), without adding demonstrations, losses, or architectural changes.

On Calvin, this augmentation increases average task completion length by *3.81%*, narrows the anchor–wrist gap by **42.4%**, and improves full five-task completion by **5%**. These results show that wrist-view generation provides effective supervisory signals and yields measurable VLA gains without extra data collection.

Similarly, on Franka-Panda demonstrations (Tab. 3), generated wrist views consistently improve task performance, confirming that the synthetic data is realistic and beneficial. Overall, wrist-view generation enriches robot datasets and yields measurable VLA gains.

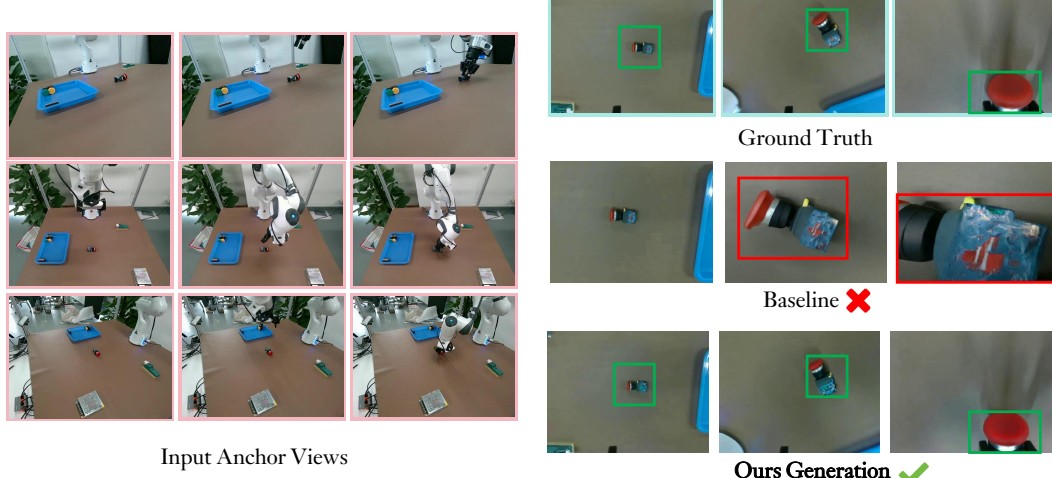

Input Anchor Views

Ground Truth

Baseline ✖

**Ours Generation** ✔

Figure 6: **Visualization on Franka real-robot data.** Multiple input anchor views (left) are used to generate wrist-view sequences (top right), which are compared with ground-truth wrist observations (bottom right). Our approach yields highly consistent predictions that closely match real data, demonstrating strong generalization from third-view to wrist-view perspectives.

| Method | Input | FVD↓ | LPIPS↓ | SSIM↑ | PSNR↑ |
|---|---|---|---|---|---|
| Ours | Left, right, top view videos | 231.43 | 0.33 | 0.78 | 17.84 |
| Ours | Left view videos | 234.30 | 0.34 | 0.78 | 18.13 |
| Ours(Zero-shot) | Left view videos | 475.55 | 0.55 | 0.70 | 14.87 |
| Cosmos | Wrist view first frame | 1156.69 | 0.65 | 0.67 | 12.59 |
| Wan 14B | Wrist view first frame | 985.99 | 0.59 | 0.68 | 13.93 |
| Ours + Cosmos | Left view first frame | 467.19 ↓689.50 | 0.58 ↓0.07 | 0.70 ↑0.03 | 14.66 ↑2.07 |
| Ours + Wan 14B | Left view first frame | **455.57** ↓530.42 | **0.57** ↓0.02 | **0.71** ↑0.03 | **14.60** ↑0.67 |

Table 5: **Plug-and-play extension to single-view world models.** Our framework enhances models trained solely on external viewpoints by synthesizing virtual wrist-view videos. We adopt Cosmos-Predict2 (NVIDIA, 2025) as the baseline Single-view World Model(SWM) and observe substantial gains. This plug-and-play design improves spatial consistency and perceptual quality across different baselines, while still delivering high-quality results with fewer anchor views. We also include a zero-shot setting, where a model trained on three views is directly tested on a single view, revealing its inherent cross-view generalization ability.

## 4.4 PLUG-AND-PLAY EXTENSION TO SINGLE-VIEW WORLD MODELS

Our framework serves as a *plug-and-play* add-on to an existing single-view world model (SVWM) while leaving the SVWM unchanged. In the baseline, the SVWM takes an *anchor-view first frame* and predicts an *anchor-view* rollout; with our module, that rollout is then converted into a temporally aligned *wrist-view* video without wrist first frame required. This post-hoc wrist synthesis resolves the core hurdle for multi-view extension, namely cross-view consistency without wrist initialization, while enriching the observation space without additional data. As shown in Tab. 5, integrating our module with Wan2.1 improves spatial and perceptual fidelity, and the gains persists.

## 4.5 ABLATION STUDY

We conduct an ablation study to disentangle the contribution of different components, as summarized in Tab. 4. The results show that the *wrist view projection* plays the most critical role: removing it leads to a drastic drop in video quality, as reflected by a large increase in FVD and degraded perceptual metrics. Moreover, the SPC loss proves essential for ensuring that the condition map carries accurate guidance information; without it, the model struggles to align wrist-view synthesis with external observations. The combination of condition map, external-view embeddings, and track-

| Training Data Size | FVD↓ | LPIPS↓ | SSIM↑ | PSNR↑ |
|:---:|:---:|:---:|:---:|:---:|
| 1k | 672.32 | 0.44 | 0.62 | 14.20 |
| 5k | 595.41 | 0.39 | 0.65 | 15.31 |
| 10k | **421.10** | **0.39** | **0.64** | **14.78** |

Table 6: Ablation on different amounts of Droid training data.

ing loss yields the best performance across all metrics. Although experiments without projection embeddings are still in progress, we estimate that their performance will be similarly poor, further reinforcing the necessity of projection-based conditioning for coherent multi-view generation.

In addition, we perform an ablation on the amount of Droid training data. The results (Tab. 6) indicate a clear scaling trend: model performance consistently improves as more data is used for training, with substantial gains from 1k to 5k samples and further improvements at 10k. This demonstrates that our approach is highly data-scalable, benefiting directly from larger and more diverse training sets.

## 5 CONCLUSION

In this work, we introduced **WristWorld**, a two-stage 4D framework for synthesizing geometrically and temporally consistent wrist-view videos from anchor-view inputs. In the reconstruction stage, a wrist head and SPC loss augment a geometric transformer to estimate poses and generate condition maps without explicit wrist supervision. These maps are then fused with CLIP semantics and text guidance in a diffusion transformer, producing high-fidelity wrist-view sequences aligned with geometry and task semantics.

Experiments on Calvin, Droid, and Franka Panda validate our framework: **WristWorld** achieves strong video generation across metrics and visual quality, while the synthesized wrist-view videos significantly boost downstream VLA learning. By augmenting datasets with wrist-centric views, **WristWorld** bridges the exocentric–egocentric gap and provides a scalable, data-driven solution for robotic training.

A current limitation of our system is the scale of training data. Due to the computational cost of training, we trained **WristWorld** on a 10k subset of Droid, which constrains its generalization breadth. As a result, the generated wrist-view sequences may occasionally exhibit minor hallucinations or inconsistencies in challenging cases. We expect these issues to diminish with larger-scale training, and scaling the data is a promising direction for further improving robustness and coverage.

**Ethics Statement**    We affirm that our work is consistent with the ICLR Code of Ethics. In conducting this research, we considered potential ethical issues such as (i) privacy and data handling, (ii) bias and fairness, (iii) misuse or dual-use risk, and (iv) conflicts of interest. For data that include human-associated content, we apply anonymization / de-identification and obtain appropriate permissions or IRB approval as necessary. We analyze bias across relevant demographic groups and report any limitations. Any sponsors or funding sources are disclosed, and we take responsibility for any potential misuse of the methods.

**Reproducibility Statement**    We have aimed to ensure that our experimental results are reproducible. The key implementation code are provided in the supplementary material. Architectural and algorithmic details are fully described in the main text and Appendix.

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

# A  APPENDIX

## A.1  USE OF LLMs

LLMs were used during coding and debugging to provide occasional technical guidance, and later to help polish the language and presentation of the manuscript.

## A.2  DATASETS

**Calvin.**  The Calvin benchmark (Mees et al., 2022) (*Composing Actions from Language and Vision*) is a simulated environment designed for studying long-horizon, language-conditioned robotic manipulation. It defines a set of five core manipulation tasks—*opening and closing drawers, switching lights, placing objects in containers, pushing blocks, and rotating knobs*—which can be composed into multi-step instructions. Each demonstration is collected via teleoperation in simulation and includes synchronized RGB observations from multiple static cameras, robot states, low-level actions, and the corresponding language command. The benchmark further provides different environment variations, enabling controlled evaluations of generalization to new objects, layouts, and task compositions. CALVIN thus serves as a standardized testbed to validate whether synthesized wrist-view videos can enhance task-conditioned learning in a controlled simulation setting.

**Droid.**  The Droid dataset (Khazatsky et al., 2024) (*Distributed Robot Interaction Dataset*) is one of the largest collections of real-world robotic demonstrations. It contains approximately 76k trajectories spanning 350 hours, collected across 564 unique scenes and 86 manipulation tasks by over 50 contributors from multiple institutions. To ensure consistency, each robot setup follows a unified configuration: a Franka Panda 7-DoF manipulator, two ZED 2 stereo cameras for external anchor views, and a ZED Mini wrist-mounted camera for egocentric observations, along with standardized calibration. Each trajectory records synchronized multi-view RGB-D streams, camera intrinsics and extrinsics, joint states, and control commands. The scale and diversity of DROID make it a challenging but rich source of anchor-view data, while the presence of a dedicated wrist camera offers an opportunity to validate generated views against ground-truth egocentric observations.

**Franka Panda demonstrations.**  To further evaluate our method on a controlled real-robot setup, we collected a dataset of $\sim$1.7 k demonstrations using a Franka Panda manipulator. The hardware setup mirrors that of DROID, with multiple fixed anchor cameras and a wrist-mounted camera, but data collection was conducted entirely in-house. Demonstrations span diverse manipulation skills such as grasping, transporting, and placing objects under natural occlusions from the robot arm. For each sequence, we log synchronized anchor-view and wrist-view videos, as well as proprioceptive states and control commands. While smaller in scale than DROID, this dataset captures calibration imperfections and actuation variability, making it particularly valuable for fine-tuning and validating wrist-view generation under true physical dynamics.

## A.3  TRAINING AND IMPLEMENTATION DETAILS

This section provides the full setup for both stages of **WristWorld** in the 1.3B configuration, with emphasis on novel components and implementation details. We also include short explanations of key terms to aid readers unfamiliar with video generation or geometric reconstruction.

**Implementation Details.**  The **wrist head** is implemented as a lightweight transformer decoder with 3 layers, 8 attention heads per layer, and an embedding dimension of 1024. It attends to aggregated multi-view tokens from VGGT and directly regresses wrist camera extrinsics $(\mathbf{R}_w, \mathbf{T}_w)$.

The **Spatial Projection Consistency (SPC) loss** is computed using dense 2D–2D correspondences. For each 3D point $\hat{\mathbf{y}}_j$, we project it into the wrist view as

$$\mathbf{u}_w'^j = \Pi(\mathbf{K}, \mathbf{R}_w, \mathbf{T}_w, \hat{\mathbf{y}}_j),$$

where $\mathbf{K}$ is the dataset-provided intrinsics. We then split the projected points into $\mathcal{S}_{\text{front}} = \{\hat{\mathbf{y}}_j \mid z_j > 0\}$ and $\mathcal{S}_{\text{back}} = \{\hat{\mathbf{y}}_j \mid z_j < 0\}$, with $z_j$ denoting the depth value in the wrist frame. The loss is

defined as

$$\mathcal{L}_u = \frac{1}{|\mathcal{S}_{\text{front}}|} \sum_{\hat{\mathbf{y}}_j \in \mathcal{S}_{\text{front}}} \text{MSE}(\mathbf{u}_w'^j, \hat{\mathbf{u}}_w^j), \quad \mathcal{L}_{\text{depth}} = -\frac{1}{|\mathcal{S}_{\text{back}}|} \sum_{\hat{\mathbf{y}}_j \in \mathcal{S}_{\text{back}}} z_j,$$

and $\mathcal{L}_{\text{proj}} = \lambda_u \mathcal{L}_u + \lambda_{\text{depth}} \mathcal{L}_{\text{depth}}$.

For video generation, we use **wrist-view projections** as conditioning inputs. Each frame is encoded by a VAE into latents $z_c^t$ and concatenated with the noisy wrist-view latents $z_w^t$. The patch embedding of the DiT is modified from a standard $2 \times 2$ convolution, expanding input channels to 32 to accommodate concatenated latents. CLIP features from anchor views are projected from 512 dimensions into the DiT token space, while text embeddings from T5 are 4096-dimensional. Temporal and view embeddings are added to capture sequence alignment and camera identity.

**Reconstruction stage (VGGT + Wrist Head).** We adopt VGGT-1B with point, depth, and camera heads (frozen) and train a new wrist head. Training images are resized to $518 \times 518$, and intrinsics are scaled accordingly. Two anchor views (*ext1*, *ext2*) are inputs, and the wrist pose is predicted relative to *ext1*. SPC loss enforces consistency: visible points are supervised with normalized reprojection error, while back-facing points are penalized if their predicted depth is negative.

| Component | Setting |
|---|---|
| Backbone | VGGT-1B |
| Image size | $518 \times 518$ |
| Token aggregation | Attention-based |
| Wrist decoder | 3 layers, 8 heads |
| Optimizer | AdamW (wd = 0.05) |
| Learning rate | $2 \times 10^{-5}$ (cosine decay) |
| Batch size | 4 per GPU, grad accum = 3 |
| Hardware | $8 \times$ A800 GPUs |
| Training time | $\sim$12h pretrain, 6h finetune |

Table 7: Reconstruction stage hyperparameters.

**Generation stage (Video DiT).** We build on Wan 1.3B DiT, which is a diffusion transformer pretrained for text-to-video. Condition maps are encoded with a VAE and concatenated with noisy wrist-view latents. CLIP embeddings from anchor views are projected into the text space and added as pseudo tokens, together with temporal and view embeddings. A maximum of 512 conditioning tokens is used. Classifier-Free Guidance (CFG)[1] is set to 5.0.

| Component | Setting |
|---|---|
| Backbone | Wan 1.3B DiT |
| Resolution | $640 \times 480$ (latent scale 1/8) |
| Patch in-channels | Expanded to 32 (for concat latents) |
| LoRA config | Rank 4, $\alpha = 4$, targets {q,k,v,o,ffn} |
| Condition tokens | 512 total (CLIP + text + temporal/view) |
| CFG | 5.0 |
| Optimizer | AdamW, lr = $1 \times 10^{-5}$ |
| Batch size | 4 per GPU |
| Hardware | $8 \times$ A800 GPUs |
| Training time | $\sim$24h pretrain, 12h finetune |

Table 8: Generation stage hyperparameters.

## A.4 VISUALIZATION

To assess appearance fidelity and viewpoint stability under realistic manipulation, we run inference on a **Franka Panda** platform for all compared video generation methods, including ours, using

---

[1]CFG balances diversity and fidelity in diffusion sampling by mixing conditional and unconditional generations. A higher value yields sharper but less diverse outputs (Ho & Salimans, 2022).

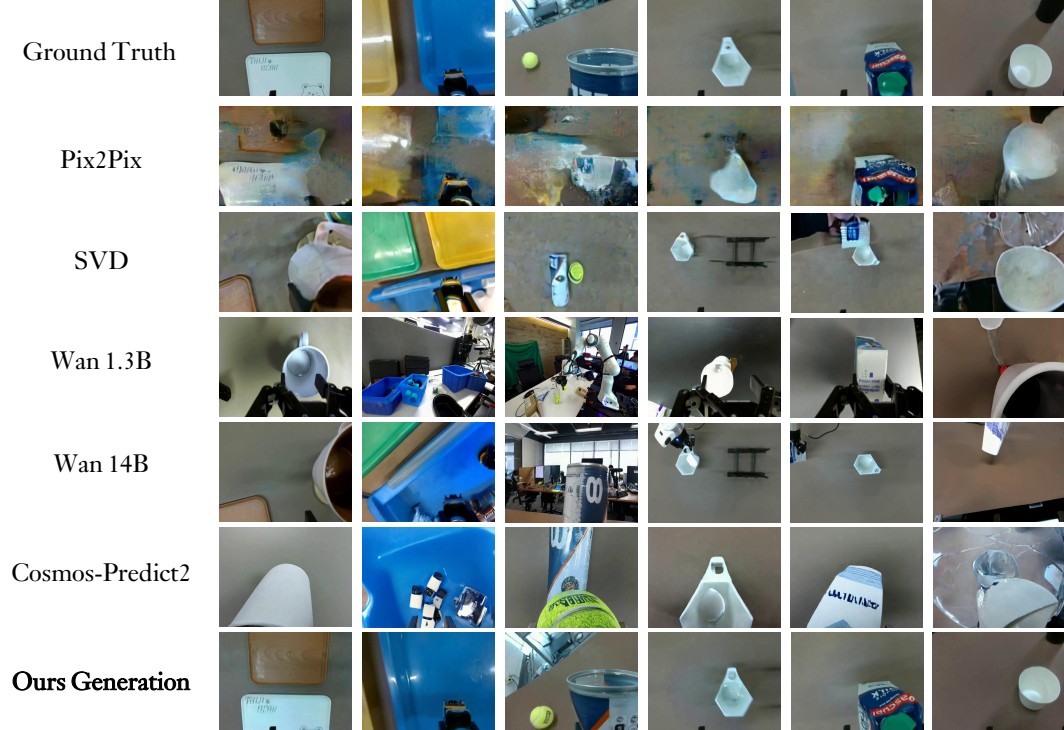

Figure 7: **Qualitative visualization on Franka Panda.** For each method, we generate videos and visualize the *middle frame* of each sequence to probe long-horizon stability. **Rows** denote methods (the top row shows ground-truth wrist-camera frames), and **columns** denote different manipulation scenes from the Franka Panda setup. Our approach maintains superior geometric consistency (crisp boundaries, coherent occlusions, stable perspective/scale) and wrist-following behavior (viewpoint motion aligned with end-effector motion and consistent object-relative poses) compared with prior models.

identical input trajectories and pre-processing. For each generated clip, we visualize the *middle frame* so as to emphasize long-horizon behavior and to minimize the bias introduced by the initial condition. The compared models comprise *Pix2Pix* (Isola et al., 2017), *SVD* (Blattmann et al., 2023), *Wan 1.3B* (Wan et al., 2025), *Wan 14B*, *Cosmos-Predict2* (NVIDIA, 2025), and **Ours**. Following common practice for controllable synthesis, **SVD**, **Wan 14B**, and **Cosmos-Predict2** are evaluated with *first-frame guidance*, while the remaining methods use their public default settings. Qualitative results are shown in Fig. 7.

Across diverse scenes, our method exhibits the strongest **geometric consistency** and **wrist-following capability**. By geometric consistency we refer to coherent object shapes and boundaries, stable perspective and scale, and physically plausible occlusions (e.g., hand–object and object–table contacts). By wrist-following we mean that the synthesized wrist-camera view remains aligned with the end-effector motion, yielding stable object-relative poses and camera parallax over time. In contrast, baselines frequently suffer from accumulated drift in the middle of the sequence: textures smear or dissolve, straight edges warp, object scale fluctuates, and the rendered viewpoint decouples from the manipulator pose, producing noticeable misalignment between the camera motion and the underlying action. Methods that rely on first-frame guidance preserve appearance early on but tend to exhibit background and pose drift as the sequence proceeds; methods without guidance avoid overfitting to the first frame but often display ghosting and fine-detail loss under fast motions or partial occlusions. While our approach may still blur very small, fast-moving details in rare cases, its overall spatial coherence and camera–motion coupling are substantially more reliable, which agrees with the improvements observed in video quantitative metrics.

## A.5 FRANKA PANDA SETTING

As shown in Figure. 8, we employ a Franka Emika Panda manipulator in our experimental setup, augmented with multiple Intel RealSense cameras to provide diverse visual perspectives. Specifically, a wrist-mounted camera enables close-up views of the manipulation workspace, while a top-mounted camera captures overhead information. In addition, left and right side cameras provide complementary viewpoints for robust perception. This multi-view sensing arrangement facilitates both accurate 3D reconstruction and reliable visual feedback for manipulation tasks.

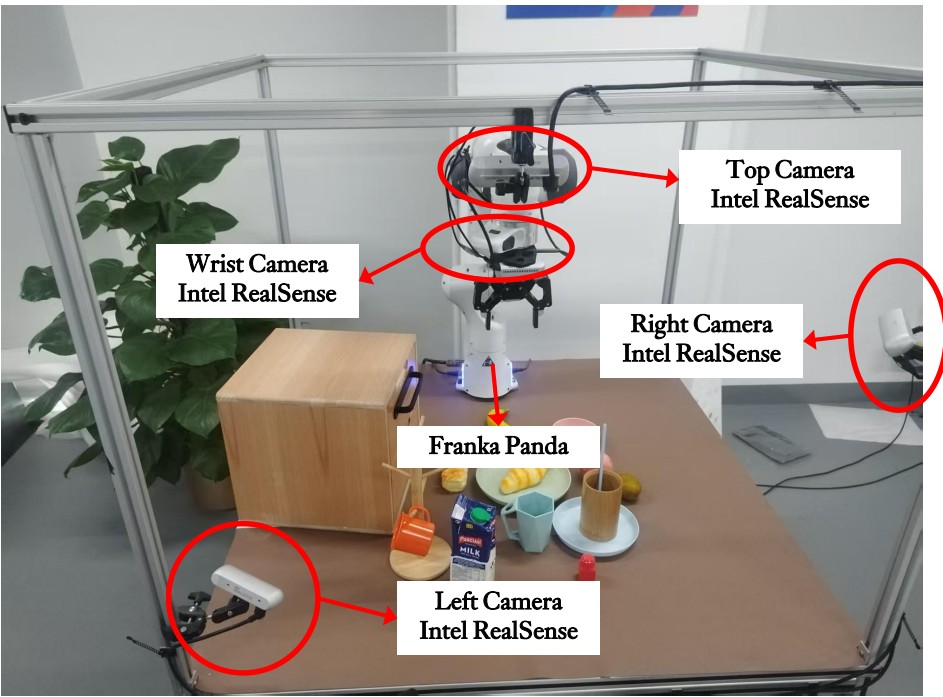

Figure 8: Experimental setup with the Franka Panda manipulator. The system is equipped with multiple Intel RealSense cameras: wrist-mounted, top, left, and right. This configuration enables multi-view visual input for robust perception and manipulation.

