# OpenReview forum: "WristWorld: Generating Wrist-Views via 4D World Models for Robotic Manipulation"
_ICLR.cc/2026/Conference — Submitted to ICLR 2026_

### Official Review · Reviewer_ZZ2e · 2025-10-19

**Soundness:** 2
**Presentation:** 3
**Contribution:** 2
**Rating:** 4
**Confidence:** 4

**Summary:**

The paper introduces WristWorld, a novel 4D world model that generates wrist-view videos from anchor (third-person) views alone—addressing a critical data scarcity problem in robotic manipulation where wrist-mounted camera data is rare but highly informative for vision–language–action (VLA) models. The method operates in two stages: (1) a reconstruction stage that extends the VGGT visual geometry model with a new wrist head and a Spatial Projection Consistency (SPC) loss to estimate geometrically consistent 4D point clouds and wrist-view camera poses without requiring ground-truth wrist supervision; and (2) a generation stage that uses a diffusion-based video model conditioned on the reconstructed projections and CLIP-encoded anchor-view semantics to synthesize temporally coherent and spatially accurate wrist-view videos.

**Strengths:**

WristWorld introduces a new solution to a real-world data bottleneck in robotics: generating high-quality wrist-view videos from only third-person (anchor) views, which are far more abundant in existing datasets. Its two-stage design—geometric reconstruction followed by diffusion-based generation—elegantly decouples pose estimation from synthesis, enabling strong spatial and temporal consistency without requiring any real wrist-view input during inference.

**Weaknesses:**

The core idea—leveraging geometric priors and diffusion models for cross-view video synthesis—builds heavily on existing frameworks (VGGT for geometry, DiT for generation) without introducing a fundamentally new algorithmic paradigm; the wrist head and SPC loss are incremental adaptations rather than conceptual breakthroughs.

Lack of direct, head-to-head comparison with the most relevant prior works in the task of anchor-to-wrist view generation, such as the "Exocentric-to-egocentric video generation" method by Liu et al. (2024), which is cited but not quantitatively benchmarked against, leaving the reader uncertain about the actual margin of improvement. The paper's central claim of being the "first" 4D world model for this task is potentially overstated, as other world models like "Tesseract" (Zhen et al., 2025) also learn 4D representations for embodiment, and the specific novelty of the two-stage pipeline is not sufficiently differentiated from this broader context.

The evaluation lacks ablation on real-world generalization: all downstream VLA gains are shown only when the synthetic wrist data is used to augment training on the same domain (e.g., Franka-trained model tested on Franka), leaving open whether the method truly generalizes across robot morphologies or environments.

**Questions:**

1.Novelty & Baselines: How does your method compare quantitatively against an adapted baseline that uses your reconstructed first wrist frame to initialize a sequence-based model like Liu et al. (2024)? This would more directly validate the need for your full geometry-based pipeline.


2.Single Anchor View: How does performance change if only one anchor view is available (common in real datasets)? Table 5 suggests minimal drop—why?


3.Significance of VLA Gains: Are the reported VLA performance improvements (e.g., +3.81% task length) statistically significant? What is the absolute performance of the true anchor+wrist upper bound? Please clarify the calculation for "closing 42.4% of the gap" to strengthen this key claim.

4.Real Policy Execution: Were VLA policies using synthetic wrist views deployed on real hardware, or only evaluated on logged data?

5.SPC Loss Supervision: The SPC loss requires 2D correspondences to the wrist view. How are these ground-truth wrist pixels obtained during training, considering the problem assumes a lack of wrist-view data? Is the SPC loss essential, or could simpler reprojection losses (e.g., using VGGT’s own depth) suffice? An ablation is missing.

6.Cross-Morphology Generalization: Does WristWorld generalize to robots with different arms/cameras, or only Franka-like setups?

---

> ### Author Response · Authors · 2025-11-20
>
> We thank the reviewer for the thoughtful assessment and for raising important questions on novelty, baselines, supervision, and generalization. We have toned down the “first 4D world model” phrasing in the Introduction. Clarifications for each point are provided below and will be incorporated into the revised paper.
>
> ---
>
> **Q1: Novelty and comparison to Exo2Ego-style methods.**
> **Response:** Exo2Ego-V relies on assumptions that do not hold in our setting: fixed multi-camera extrinsics, four external views, and an egocentric first frame at inference. Our setup has none of these. For completeness, we attempted to reproduce Exo2Ego-V on our robotic data following the official setup, but it failed to produce meaningful wrist videos and mostly generated degenerate outputs. This indicates that the method does not transfer straightforwardly to our embodied scenario. As its core assumptions are incompatible and its reproduced behavior was unusable, we did not include it as a quantitative baseline, but will position it clearly as related work with different applicability.
>
> ---
>
> **Q2: Why is single-view performance close to multi-view in Table 5?**
> **Response:** As shown in Table 5, the gap between single- and multi-view settings is small because the Franka workspace is compact, and a single anchor camera already captures most relevant objects and the arm with minimal occlusion. Additional views therefore add limited geometric information. This observation is consistent with our RealBot experiments. We will emphasize that this effect is workspace-dependent and should not be over-generalized.
>
> ---
>
> **Q3: Interpretation of the 3.81% gain and the 42.4% gap closure.**
> **Response:**
> Let
> - AvgLen(no wrist) = 3.67,
> - AvgLen(gt wrist) = 4.00,
> - AvgLen(gen wrist) = 3.81.
>
> Relative improvement over the no-wrist baseline:
>
> $$\frac{3.81 - 3.67}{3.67}=3.81\\%$$
>
> Gap closure toward ground-truth wrist performance:
>
> $$\frac{3.81 - 3.67}{4.00 - 3.67} = 42.4\\%$$
>
>
> This measures how much of the wrist-view advantage is recovered by generated wrist videos. Since synthetic wrist views cannot exceed the performance of true wrist views, gap closure is the appropriate metric. We will add this explanation in the revision.

---

> ### Author Response · Authors · 2025-12-02
>
> **Q4: Deployment of VLA policies with synthetic wrist views on real hardware.**
> Table 3 presents the real-world performance of VLA policies deployed on a physical Franka Panda arm. When only the external anchor view is provided, success rates are significantly limited due to missing wrist-level visual feedback. By supplying **our generated wrist views** during both training and deployment, the policy achieves **substantial gains across all tasks**, improving the average success rate from **37.8% → 53.3% (+15.5%)**, and in some cases recovering more than **75% of the gap** toward the full “Anchor + Wrist” setup that requires a real wrist-mounted camera. These results confirm that our synthetic wrist observations are sufficiently reliable for controlling real hardware, and directly address the reviewer’s concern regarding the practicality of deploying VLA policies with generated wrist views.
>
>
> | Inputs                     | Close the upper drawer | Pick bread and place into drawer | Pick up the milk | Mean  |
> |----------------------------|------------------------|-----------------------------------|-------------------|--------|
> | Anchor + Wrist             | 80.0%                  | 73.3%                             | 46.7%             | 66.7% |
> | Anchor                     | 60.0%                  | 40.0%                             | 13.3%             | 37.8% |
> | Anchor + Ours Gen Wrist    | 73.3% **(+13.3)**      | 53.3% **(+13.3)**                 | 33.3% **(+20.0)** | 53.3% **(+15.5)** |
>
>
> ---
>
> **Q5: How SPC is supervised without wrist data, and whether it is essential**
> **Response:**
> SPC is used only during training. We train it on datasets that contain wrist views (e.g., Droid), using VGGT to provide both 2D correspondences and predicted depth. Thus, SPC never requires real wrist depth and relies solely on VGGT outputs. After this stage, the model can be applied to datasets without wrist views (e.g., Real) without modification.
>
> Simpler reprojection losses were insufficient because they depend on accurate metric depth, which VGGT does not provide in sparse embodied setups. SPC instead enforces projection consistency directly in image space, avoiding depth errors. As shown in **Table 4 (second row)**, removing SPC causes large performance drops, confirming that it is essential.
>
> ---
>
> **Q6: Generalization of WristWorld across robot morphologies.**
> **Response:** WristWorld already generalizes across different camera layouts and from simulation to a physical Franka arm. Since it operates purely on image geometry and not robot states, it is in principle adaptable to other manipulators. Due to resource limitations we have not tested additional platforms, and will note this as a limitation and natural future direction.
>
> ---
>
> **In conclusion**, we appreciate the reviewer’s constructive feedback. We have clarified the novelty relative to Exo2Ego-style methods, the behavior under single vs. multi-view setups, the gap-closure metric, the role of SPC, and the expected generalization properties. These clarifications will be included in the revised version.

---

### Official Review · Reviewer_fkUe · 2025-10-29

**Soundness:** 3
**Presentation:** 2
**Contribution:** 3
**Rating:** 6
**Confidence:** 3

**Summary:**

This work proposes a new wrist video generation model to alleviate the scarcity of wrist video data. It achieves this by proposing a reconstruction-generation pipeline. The reconstruction part aims to reconstruct a 4D point cloud from anchor videos and wrist view camera extrinsic parameters. The imperfect wrist view video can be rendered from these extrinsic parameters. The generation part focuses on generating a perfect wrist view video from these cues.

**Strengths:**

The targeted problem is meaningful, and the solution is sound. This work proposes to upgrade VGGT with an extra head to predict wrist view camera extrinsics. To do so, this work introduces a correspondence supervision loss, referred to as spatial projection consistency loss. Afterward, this work injects the rendered imperfect wrist view video and anchor video into a video diffusion model to generate a perfect wrist video.

**Weaknesses:**

The presentation can be improved. I notice that the generated wrist view videos are repeated multiple times in your provided website (intro video and visualization part) as well as in the main manuscript. It is beneficial to showcase qualitative results as extensively as possible. Moreover, it is better to display the anchor video alongside the generated video. Furthermore,  the explanation is unclear:
1. The used training data is unclear. What did you use for the first stage of reconstruction and generation training? And for the second stage? Why was only 10k Droid data used? What about more Droid data?
2. L150: S is missing an explanation.
3. The subscript of y in lines 213 and 214 is different.
4. The depth term of SPC is not very clear. Why is only the back term supervised? What is the rationale behind this loss design?
5. How do you obtain correspondence points? What algorithm do you use?
6. What does "the i-th external view" mean in L261?
7. There is no explanation about "cross-view fine-tuning." What does it mean? The training strategy seems novel; why don't you provide more details?
8. What video generation model do you use? Is it trained from scratch in a newly designed network?
9. L426:  "persis"?

**Questions:**

1. The used training data is unclear. What did you use for the first stage of reconstruction and generation training? And for the second stage? Why was only 10k Droid data used? What about more Droid data?
2. L150: S is missing an explanation.
3. The subscript of y in lines 213 and 214 is different.
4. The depth term of SPC is not very clear. Why is only the back term supervised? What is the rationale behind this loss design?
5. How do you obtain correspondence points? What algorithm do you use?
6. What does "the i-th external view" mean in L261?
7. There is no explanation about "cross-view fine-tuning." What does it mean? The training strategy seems novel; why don't you provide more details?
8. What video generation model do you use? Is it trained from scratch in a newly designed network?

---

> ### Author Response · Authors · 2025-11-20
>
> We sincerely thank the reviewer for the detailed and constructive feedback. We have addressed all presentation issues, including missing definitions and notation corrections, and we are preparing a data-scaling experiment (1k and 5k Droid subsets) to further illustrate how performance changes with larger training sets.
>
> **Q1: Clarification on the training data and the use of 10k Droid videos**
> **Response:**
> Thank you for raising this question. Both the reconstruction module and the video generation module are pretrained on the Droid dataset, followed by cross-view finetuning on Franka. The choice of 10k videos is due to the computational cost of VGGT reconstruction and video diffusion training, which limits the feasible dataset size. We additionally include experiments on 1k and 5k Droid subsets (Table 6), which clearly reveal a consistent scaling trend, and the method continues to improve as more pretraining data is used.
>
> | Training Data Size | FVD↓   | LPIPS↓ | SSIM↑ | PSNR↑ |
> |--------------------|--------|--------|--------|--------|
> | 1k                 | 672.32 | 0.44   | 0.62   | 14.20  |
> | 5k                 | 595.41 | 0.39   | 0.65   | 15.31  |
> | **10k**            | **421.10** | **0.39** | **0.64** | **14.78** |
>
>
> **Q2: Missing explanation of S in L150**
> **Response:**
> We appreciate you pointing this out. We have added an explicit definition of S at its first appearance, clarifying that it denotes the correspondence set used for projection consistency. (See L151)
>
> **Q3: Subscript inconsistency in lines 213–214**
> **Response:**
> Thank you for catching this. The notation has been corrected for consistency. (See L216)
>
> **Q4: Clarification on the depth term in SPC and why only the “back” region is supervised**
> **Response:**
> The depth term in SPC is not designed to regress absolute depth. Its purpose is to avoid the physically invalid case where projected points fall behind the camera. Wrist-view depth is not available, and RealSense depth contains substantial noise and many missing pixels, making it unreliable for supervising front-depth. Negative depth values, on the other hand, are always invalid and therefore provide a stable geometric constraint. For this reason, we penalize only the back region through the −zⱼ term to ensure projections remain physically valid.
>
> **Q5: How correspondence points are obtained**
> **Response:**
> The correspondence points come from the 2D matching head of VGGT. All anchor views (and the ground-truth wrist view during precomputation) are processed by VGGT to obtain 2D–2D correspondences, which are then lifted into 3D using the reconstructed point cloud before being used for SPC computation.
>
> **Q6: Clarification of the phrase “the i-th external view”**
> **Response:**
> Thank you for noting this ambiguity. The phrase simply refers to indexing multiple anchor views, such as left, right, or top. We have revised the wording to “the i-th anchor view” for clarity. (See L267)
>
> **Q7: Missing explanation of cross-view finetuning**
> **Response:**
> Thank you for pointing this out. Cross-view finetuning is required when moving from two-view pretraining on Droid to three-view finetuning on Franka. The video generation model includes a view-embedding module whose dimensionality depends on the number of anchor views. During finetuning, we load all pretrained weights except the view-embedding matrix, which is randomly initialized and trained jointly. This explanation has now been added. (See L356–L359)
>
> **Q8: Video generation model choice and training procedure**
> **Response:**
> We use Wan2.1 1.3B[1] as the video diffusion backbone. We do not train it from scratch; instead, we finetune the publicly released pretrained weights. Wan2.1 1.3B provides strong generation quality, long temporal support (81 frames), and a model size that is feasible to train within our setup.
>
> **In conclusion**, we thank the reviewer once again for the thoughtful feedback. We have clarified the technical details, addressed presentation issues, and are preparing additional data-scaling experiments to further strengthen the paper.
>
> *Refs:*
> [1] Wan: Open and Advanced Large-Scale Video Generative Models. arXiv 2025

---

### Official Review · Reviewer_urB1 · 2025-11-01

**Soundness:** 4
**Presentation:** 4
**Contribution:** 3
**Rating:** 8
**Confidence:** 3

**Summary:**

The paper introduces WristWorld, a two-stage framework designed to synthesize wrist-view videos using only third-person (anchor) views, without requiring a wrist first-frame. Stage 1 (Reconstruction) extends VGGT with a wrist head that predicts wrist camera extrinsics and introduces a Spatial Projection Consistency (SPC) loss to enforce alignment between RGB correspondences and 3D/4D geometry. Stage 2 (Generation) employs a diffusion transformer conditioned on (i) wrist-view projections and (ii) CLIP features derived from third-person views and text instructions, enabling the generation of temporally coherent wrist-view videos. Evaluations on the DROID and Franka datasets demonstrate that WristWorld outperforms baselines across four video quality metrics and improves downstream Visual Language Action (VLA) performance on CALVIN.

**Strengths:**

- The authors propose a simple yet effective framework for synthesizing wrist-camera views from third-person perspectives.
- The generated videos can directly benefit downstream models such as Visual Language Action (VLA) models.
- The paper presents extensive experimental benchmarks conducted in both real-world and simulated environments.

**Weaknesses:**

- When examining the 3D baseline without the SPC loss, there is a noticeable difference between the ground truth and the estimated point cloud. Even after applying the SPC loss, a small discrepancy remains. Is this error primarily due to inaccuracies in the estimated point cloud or in the predicted wrist pose? It would be helpful to report the wrist pose error and provide a qualitative comparison using the ground-truth wrist pose, showing both the estimated and ground-truth point clouds.
- How sensitive are the results to the number of cameras used? Can the proposed method achieve comparable performance when the number of cameras is reduced to one or two?

**Questions:**

See weaknesses.

---

> ### Author Response · Authors · 2025-11-20
>
> Thank you for the constructive and insightful comments. We address both points below with additional clarification.
>
> **Q1: Residual mismatch between ground-truth and predicted geometry**
> **Response:** We appreciate the reviewer highlighting this issue, as it directly connects to the motivation for introducing the **SPC Loss**. While VGGT generates point clouds with reasonable relative geometry, their absolute depth remains unreliable. Supervising wrist pose using ground-truth extrinsics forces the model to rely on these imperfect depths, causing errors to accumulate during wrist-view projection. We also attempted to incorporate RealSense depth as supervision, but its measurements contain substantial noise and frequent missing regions, making it unsuitable for stable learning. Since the final wrist-view is produced through projection, we instead guide the model from the **projection viewpoint**, enforcing cross-view agreement directly in image space. This sidesteps the need for accurate metric depth and leads to significantly more reliable wrist-pose estimation. Consistent with this reasoning, **Table 4** shows that replacing SPC Loss with ground-truth pose supervision (second row) results in clear degradation across all metrics.
>
> | SPC Loss | FVD ↓  | LPIPS ↓ | SSIM ↑ | PSNR ↑ |
> |----------|--------|---------|--------|---------|
> | ✗         | 790.10 | 0.59    | 0.47   | 10.75   |
>  | **✓**        | **421.10** | **0.39**    | **0.64**   | **14.78**   |
>
> ---
>
> **Q2: Sensitivity to fewer camera views**
> **Response:** As shown in the first two rows of **Table 5**, reducing anchor views from “left, right, top” to a single “left” view produces only a modest drop in performance. This is aligned with our RealBot findings: a single external camera already observes the workspace and arm configuration with minimal occlusion, enabling the reconstruction pipeline to function effectively. Consequently, the model remains robust even under sparse viewing conditions, and the impact of removing additional views is limited in this setting.
> Beyond the fully supervised settings, we also include a **zero-shot evaluation** (third row of Table 5), where the model receives only the left-view videos but no paired wrist-view supervision at all. As expected, the absence of supervision results in larger errors, yet the system still produces plausible wrist-view predictions. This further validates that our projection-driven architecture generalizes reasonably even without explicit wrist-view training, demonstrating robustness under highly sparse and weakly supervised conditions.
>
> | Method            | Input                        | FVD↓   | LPIPS↓ | SSIM↑ | PSNR↑ |
> |-------------------|------------------------------|--------|--------|--------|--------|
> | Ours              | Left, right, top view videos | 231.43 | 0.33   | 0.78   | 17.84 |
> | Ours              | Left view videos             | 234.30 | 0.34   | 0.78   | 18.13 |
> | Ours (Zero-shot)  | Left view videos             | 475.55 | 0.55   | 0.70   | 14.87 |
>
> ---
>
> **In conclusion**, SPC Loss is crucial because it supervises wrist-pose estimation through projection consistency rather than unreliable depth, and the framework retains strong performance even when operating with fewer anchor views.

---

### Official Review · Reviewer_oWxd · 2025-11-01

**Soundness:** 3
**Presentation:** 2
**Contribution:** 1
**Rating:** 2
**Confidence:** 5

**Summary:**

This paper proposes a framework to generate wrist-view robot manipulation videos, given the input of anchor-view robot videos. To realize this, the anchor-view videos are reconstructed by VGGT and projected to wrist view at the first stage. Next, a video generation model is utilized to fully synthesize wrist-view videos. The evaluation is conducted on several datasets.

**Strengths:**

1. The paper is well-organized.
2. The focus on wrist-view manipulation is meaningful, and the evaluation shows that wrist-view data is important for robot tasks.

**Weaknesses:**

1. The novelty and the contribution are **very limited**, which is not enough for the ICLR bar:
(1) The proposed method is substantially a *novel-view video* synthesized pipeline; such a "reconstruction-render/project-videogen" idea has been fully investigated in many existing 3D works, such as [1, 2, 3]. This work just utilizes a similar idea to the robot domain, where no specific challenges are discussed or solved.
(2) The statement of "world model" is *significantly over-claimed*. World-model denotes the ability to predict the state/action of the future, given only the *previous* observation, instead of the complete observation of finishing a task. Further, "4D" World Model is *more over-claimed*, as *NO* 4D prediction is introduced, but only the 2D novel-view video is synthesized.
(3) The pipeline needs the complete manipulation videos of anchor-view. However, such input conditions are not easy to gain for a robot manipulation task. A natural question is: When the complete manipulation video, although from a different view, is given, **the whole information about the "task completion" (which is more important for the robot) *has been provided***, why still need the wrist-view to complete the task?
2. No limitation or failure case is provided.

*Refs*:
[1] GEN3C: 3D-Informed World-Consistent Video Generation with Precise Camera Control. CVPR 2025.
[2]  You See it, You Got it: Learning 3D Creation on Pose-Free Videos at Scale. CVPR 2025.
[3] SpatialCrafter: Unleashing the Imagination of Video Diffusion Models for Scene Reconstruction from Limited Observations. CVPR 2025.

**Questions:**

The work may be a good project, but definitely not a very insightful research paper, at least for ICLR or top-tier venues.

---

> ### Author Response · Authors · 2025-11-20
>
> We sincerely thank the reviewer for the constructive feedback. We respectfully point out that the core contribution of our work lies not in generic novel-view video synthesis, but in solving a long-standing and uniquely embodied challenge: generating *cross-view consistent wrist-view observations from extremely sparse and discrete camera setups*, a setting where existing 3D or view-synthesis methods fundamentally fail. Our method directly addresses this gap through our SPC Loss and cross-view reasoning design, and importantly provides **substantial, measurable gains** for VLA models. This demonstrates that our approach is not a simple novel-view pipeline, but a practically impactful solution to a critical bottleneck in embodied robot learning.
>
>
> ---
>
> **Q1: Limited novelty; similar to a novel-view pipeline.**
> **Response:**
> Thank you for the question. The key novelty of our work is solving a challenge **specific to embodied intelligence** that existing 3D or novel-view pipelines cannot handle: generating wrist-view observations under **extremely sparse and highly discrete camera setups**, where the wrist camera pose is *unknown* and must be inferred from a single anchor view. Prior works such as GEN3C, You See It You Got It, and SpatialCrafter assume dense views or predefined trajectories and therefore do **not** address this wrist-pose estimation problem.
>
> | Method                    | Dynamic Target Pose                       | Use for Data Engine |
> |---------------------------|---------------------------------------------|----------------------|
> | Previous Novel-view Methods | ✗ (fixed poses or predefined trajectories) | ✗                    |
> | **Ours**                  | ✓ (estimate wrist pose from sparse anchor view) | ✓                    |
>
>
> Our SPC Loss is designed exactly for this setting and significantly improves wrist-pose estimation, as shown in Table 4. More importantly, our goal is not generic novel-view synthesis but **multi-view consistency for embodied world models**, which directly leads to substantial VLA performance gains that existing methods cannot achieve (Table 2 & 3).
>
> | Method           | Inputs      | 1/5  | 2/5  | 3/5  | 4/5  | 5/5  | Avg. Len. |
> |------------------|-------------|------|------|------|------|------|-----------|
> | VPP (GT Wrist. Upper-bound) | Static RGB + Gripper RGB | 94.9% | 86.8% | 80.4% | 72.9% | 65.4% | 4.00 |
> | VPP (No Wrist baseline) | Static RGB | 91.2% | 82.2% | 73.2% | 65.2% | 55.4% | 3.67 |
> | **VPP + Ours (Ours Gen Wrist)** | **Static RGB** | **92.9%** | **84.2%** | **75.4%** | **67.6%** | **60.4%** | **3.81** |
>
>
> ---
>
> **Q2: Over-claimed use of “world model.”**
> **Response:**
> We understand the reviewer’s concern and clarify our usage. In recent literature, world models are not limited to temporal prediction of future states or actions. Models that perform **spatial inference, imagination** are also widely regarded as world models ([2][3][4]]). Our system fits this broader category by enabling the generation of multi-view consistent videos grounded in 3D structure. Furthermore, our model is designed to be **zero-shot plug-and-play compatible** with predictive world models. Current single-view world models (e.g., Cosmos) lack spatial imagination and cannot generate multi-view consistent predictions. Our method enables such models to produce spatially aligned multi-view outputs and even reconstruct temporally evolving point clouds, as shown in our demo videos. Thus, our approach expands the functional capacity of single-view world models rather than merely generating videos.
>
> ---
>
>
> **References:**
> [1] EnerVerse: Envisioning Embodied Future Space for Robotics Manipulation
> [2] Physically Embodied Gaussian Splatting: A Realtime Correctable World Model for Robotics
> [3] Cosmos-Drive-Dreams: Scalable Synthetic Driving Data Generation with World Foundation Models
> [4] Street-View Image Generation from a Bird's-Eye View Layout

---

> > ### Author Response · Authors · 2025-12-02
> >
> > **Q3: Why need wrist-view data if the full anchor-view video is already given?**
> > **Response:**
> > We clarify two important points. First, our method does **not** require full task-completion sequences. Partial clips of trajectories are sufficient, and the generated wrist-view sequence does not assume the existence of a complete execution. Second, our pipeline is designed as a **data-augmentation paradigm** for training VLA models, where full anchor-view videos are readily available in the dataset. The need for wrist-view generation is motivated by empirical evidence: wrist-view observations dramatically improve VLA performance. This is supported by our VLA experiments (Tables 2 and 3), which show that wrist-view input substantially boosts policy success rates. Therefore, our contribution is practical and meaningful: enabling wrist-view augmentation for datasets where collecting real wrist-view footage is difficult or impossible.
> >
> > ---
> >
> > **Q4: Missing limitations and failure cases.**
> > **Response:**
> > Thank you for pointing this out. Our method can exhibit occasional hallucinations or fine-scale inconsistencies, largely due to the limited training data available under the computational budget. We are preparing an additional experiment showing that increased data volume reduces these issues. A concise discussion of this limitation has been added in the Conclusion (see Conclusion, third paragraph).
> >
> > ---
> >
> > **In conclusion**, we again thank the reviewer for the insightful feedback. We have clarified the core novelty of addressing multi-view sparsity in embodied intelligence, the intended meaning of our world-model formulation, the necessity of wrist-view generation for improving VLA performance, and the current limitations. We hope these clarifications resolve the concerns and more accurately convey the contributions of our work.

---

### Author Response · Authors · 2025-11-27
**Update of PDF & Point-by-Point Reply**

We sincerely thank all reviewers for the detailed, constructive, and time-consuming feedback. We have carefully revised the paper, clarified the method, added new experiments, and provided detailed point-by-point replies under each reviewer’s section.

The updates in the revised PDF (highlighted in red) address concerns raised by the reviewers across four key dimensions:

---

## **1. Method Clarification & Technical Corrections**
These updates were made in direct response to concerns raised regarding missing definitions, symbol inconsistencies, and unclear terminology.

### **Typo fixes and symbol corrections**
- To address `Reviewer fkUe`’s concerns:
  - We corrected inconsistent notation and subscripts (L216–217).
  - We added the missing explanation of *S* (L151).
  - We replaced the ambiguous phrase **“the i-th external view”** with **“the i-th anchor view”** (L267).
  - Corrected spelling errors such as *“persis”* (L477).

### **Training data clarification**
- To address `Reviewer fkUe` and `Reviewer ZZ2e`:
  - We clarified the training datasets used for both reconstruction and generation modules.
  - We explained why 10k Droid samples were used and described computational constraints (L497–501).

---

## **2. Additional Explanation of Key Components**
Reviewers requested deeper descriptions of important procedures and loss terms.

### **Cross-view finetuning description**
- To address `Reviewer fkUe`:
  - We added a full explanation of the cross-view finetuning strategy, including view-embedding reinitialization when moving from 2-view Droid training to 3-view Franka finetuning (L356–L359).

### **Limitation & failure-case discussion**
- To address `Reviewer oWxd` and `Reviewer ZZ2e`:
  - We added explicit discussion of limitations and observed failure modes in the Conclusion, including occasional hallucinations and fine-scale inconsistencies.

### **Clarification on data usage and Droid-10k choice**
- To address `Reviewer fkUe` and `Reviewer ZZ2e`:
  - We further clarified the selection of Droid subsets and the role of precomputation cost.

---

## **3. New Experiments & Expanded Evaluation**
Several reviewers requested additional experimental evidence concerning scalability, sensitivity, and baselines.

### **Data-scaling experiments (Table 6)**
- To address `Reviewer fkUe` and `Reviewer ZZ2e`:
  - We added new experiments using **1k** and **5k** Droid subsets.
  - Results show clear upward scaling trends as data increases (L486–499).

### **Zero-shot view-number transfer (Table 5 update)**
- To address `Reviewer urB1` and `Reviewer ZZ2e`:
  - We added results comparing **zeroshot 1-view** settings to evaluate camera-count sensitivity.

---

### Meta-Review · Area_Chair_puyP · 2026-01-02

**Summary:**

This paper studies wrist-view generation from anchor views using a two-stage reconstruction and video generation pipeline. Reviewers agree the problem is relevant and the system is functional, but raise consistent concerns about limited novelty, heavy dependence on existing geometry and video models, and over-claimed framing as a “4D world model.” Despite clarifications and added experiments in the rebuttal, the overall contribution remains below the ICLR bar. The AC recommends rejection.

**Reviewer Concerns:**

The main concerns relate to contribution and positioning rather than correctness. Reviewers question whether the method goes beyond adapting existing novel-view reconstruction and video synthesis pipelines, and whether the claimed “world model” contribution is justified. Comparisons to the most relevant prior work are viewed as incomplete, and the necessity of the full pipeline is not convincingly isolated. Although the rebuttal clarifies terminology, adds experiments, and addresses presentation issues, the core concerns about novelty, framing, and comparative strength remain.

**Reviewer Scores:**

- Reviewer oWxd:  2, citing limited novelty and over-claimed contributions. This assessment would remain unchanged.
- Reviewer fkUe:  6, noting solid engineering but reservations about novelty and evaluation scope. After the rebuttal, the score would remain borderline.
- Reviewer urB1: 8, emphasizing empirical performance and practical relevance.
- Reviewer ZZ2e: 4, raising concerns about novelty, baselines, and generalization. These concerns persist after rebuttal.

---

### Decision · Program_Chairs · 2026-01-26

Reject